# PFKFB4 interacts with ICMT and activates RAS/AKT signaling-dependent cell migration in melanoma

Méghane Sittewelle[1,2], Vincent Kappès[1,2], Chenxi Zhou[1,2], Déborah Lécuyer[1,2], Anne H Monsoro-Burq[1,2]

**Cell migration is a complex process, tightly regulated during embryonic development and abnormally activated during cancer metastasis. RAS-dependent signaling is a major nexus controlling essential cell parameters including proliferation, survival, and migration, utilizing downstream effectors such as the PI3K/AKT signaling pathway. In melanoma, oncogenic mutations frequently enhance RAS, PI3K/AKT, or MAP kinase signaling and trigger other cancer hallmarks among which the activation of metabolism regulators. PFKFB4 is one of these critical regulators of glycolysis and of the Warburg effect. Here, however, we explore a novel function of PFKFB4 in melanoma cell migration. We find that PFKFB4 interacts with ICMT, a posttranslational modifier of RAS. PFKFB4 promotes ICMT/RAS interaction, controls RAS localization at the plasma membrane, activates AKT signaling and enhances cell migration. We thus provide evidence of a novel and glycolysis-independent function of PFKFB4 in human cancer cells. This unconventional activity links the metabolic regulator PFKFB4 to RAS-AKT signaling and impacts melanoma cell migration.**

## Introduction

Cell migration is one of the critical processes involved in the formation and maintenance of multicellular organisms. To acquire motility, cells activate complex properties such as cytoskeleton remodeling, inhibition of cell–cell contacts, remodeling the extracellular matrix and response to chemo-attractants (Roussos et al, 2011; Theveneau & Mayor, 2014). To orchestrate these processes, numerous redundant and complementary signaling pathways cooperate with one another. Although many of these pathways have been studied separately, the crosstalk between signaling and other parameters, such as cell cycle or cell metabolism, only begins to be explored in stem cells, normal development, and cancer (Hanahan & Weinberg, 2011). Cell motility and invasiveness properties are reactivated during cancer progression following the aberrant activation of multiple cellular programs, such as growth factor–independent signaling, metabolic, and epigenetic reprograming, which cooperate to sustain growth, proliferation, and survival properties in the primary tumor (Hanahan & Weinberg, 2011). Cancer cell migration, metastasis and formation of secondary tumors are the major cause of death for aggressive cancers such as cutaneous melanoma, the deadliest skin cancer in humans (Bertolotto, 2013). The mechanisms driving melanoma cell invasion are multiple and remain incompletely understood. To treat melanoma, one possible therapeutic strategy is to target identified driver mutations. The numerous genetic alterations involved in cutaneous melanoma development can be classified into four subtypes: BRAF, RAS (N/H/K), NF1, and Triple-WT (Cancer Genome Atlas Network, 2015). The MAPK pathway is the most frequently altered with ~50% of tumors mutated in *BRAF* gene, followed by 25% with mutations in *NRAS* and 14% in *NF1*, leaving 10% of the tumors with no identified driver mutation and more complex genetic landscapes (Ali et al, 2013; Cancer Genome Atlas Network, 2015). Although RAS signaling acts upstream of both MAPK and PI3K/AKT signaling (Cox & Der, 2010), it is interesting to note that the two main *NRAS* activating mutations in cutaneous melanoma, the hotspots Q61 (80% of *NRAS* subtype tumors) and G12 (15%), drive a differential activation of the downstream pathways with preferential activation of MAPK or PI3K/AKT, respectively, suggesting a complex modulation of the structure and activity of oncogenic proteins (Posch et al, 2016). In all cases, these alterations in signaling lead to increased melanoma cell proliferation, survival and migration.

In addition to mutations activating signaling pathways, metabolism rewiring allows cancer cells to promote active cell proliferation. In particular, enhanced glycolysis rate even under normal oxygen conditions, called the Warburg effect, drives many parallel biosynthesis pathways to provide cellular building blocks together with bioenergy (Liberti & Locasale, 2016). Moreover, the hypoxic environment often found in early primary tumors before vascularization, also stimulates the activation of metabolic regulators induced by the hypoxia-inducing factor HIF1. This is the case for the family of 6-phosphofructo-2-kinase/fructose-2,6-biphosphatases enzymes (PFKFB1-4), which are major regulators of glycolysis, controlling the rate of the second irreversible and rate-limiting reaction of glycolysis catalyzed by the phosphofructokinase 1 (PFK1) (Hers & Van Schaftingen, 1982). PFKFB enzymes are bi-functional

[1]Université Paris-Saclay, Faculté des Sciences d'Orsay, CNRS UMR 3347, INSERM U1021, Orsay, France [2]Institut Curie Research Division, PSL Research University, CNRS UMR 3347, INSERM U1021, Orsay, France

Correspondence: anne-helene.monsoro-burq@curie.fr, msittewelle@gmail.com

and synthesize (with kinase activity) or degrade (with phosphatase activity) the fructose-2,6-biphosphate, the main allosteric activator of PFK1. Thus, increased PFKFB protein kinase activity promotes glycolysis. In human, four distinct genes encode PFKFB isoenzymes 1–4, each one possessing many splicing isoforms and differing in their tissue-specific abundance, kinetics and regulation properties (van Schaftingen et al, 1982; Rider et al, 1992, 2004; Pilkis et al, 1995; Bruni et al, 1999; Manes & El-Maghrabi, 2005). PFKFB proteins are overexpressed in cancer. In particular, increased PFKFB4 levels have been reported in several human tumors, including cutaneous melanoma (Minchenko et al, 2004, 2005a, 2014; Goidts et al, 2012). Moreover, *PFKFB4* is induced by hypoxia, is required for survival and proliferation of normal thymocytes (Houddane et al, 2017) as well as of several cancer cell lines such as lung, breast, and colon ade-nocarcinomas and prostate and bladder cancer (Ros et al, 2012; Yun et al, 2012; Chesney et al, 2014; Zhang et al, 2016b). So far, most studies have focused on cell metabolism reprograming by PFKFB4 and have proposed that PFKFB4 is a major driver of Warburg effect (Minchenko et al, 2005a, 2005b, 2014; Goidts et al, 2012; Ros et al, 2012; Yun et al, 2012; Chesney et al, 2014, 2015; Shu et al, 2016; Zhang et al, 2016b; Houddane et al, 2017; Yao et al, 2019). However, a few recent studies have identified alternative functions of PFKFB4, outside of its canonical control of glycolysis. For example, PFKFB4 regulates small cell lung-cancer chemo-resistance by stimulating autophagy via its interactions with Etk tyrosine kinase (Strohecker et al, 2015; Wang et al, 2018). PFKFB4 also operates as a protein kinase and directly phosphorylates SRC-3, promoting metastatic progression in highly glycolytic breast cancer cells (Dasgupta et al, 2018). During development, PFKFB4 is essential for early embryonic induction and neural crest cells migration through the activation of AKT signaling (Pegoraro et al, 2015; Figueiredo et al, 2017). In cancer, the intriguing relationships between PFKFB4, cell signaling and cell migration remain unexplored.

Here, we have analyzed the importance of PFKFB4 in melanoma cell migration. Using human metastatic melanoma cell lines with high *PFKFB4* expression (Rambow et al, 2015), we show that PFKFB4 activity is required for active cell migration in several different cellular contexts, without a connection to the rate of glycolysis. Rather, we identify potential interacting proteins by mass spec-trometry, among which we validate the protein–protein interactions between PFKFB4 and isoprenylcystein carboxymethyl transferase (ICMT), an enzyme essential for RAS posttranslational modifications controlling its localization at the plasma membrane. Our study further defines a novel, glycolysis-independent function for PFKFB4, which promotes ICMT–RAS interactions, results in efficient RAS lo-calization at the plasma membrane, activates AKT signaling and enhances melanoma cell migration.

# Results

## PFKFB4 controls metastatic melanoma cell migration in vitro in a glycolysis-independent manner

Melanomas present higher expression of PFKFB4 mRNA compared with other tumors (Fig S1). We have previously linked elevated expression of PFKFB4 with embryonic cell migration in vivo (Figueiredo et al, 2017), but in melanoma, whereas PFKFB4 has been linked to promoting the Warburg effect, its role in cell migration remains to be explored. Here we have chosen two human mela-noma cell lines expressing high levels of PFKFB4 (MeWo and A375M, Fig 1A and B, [Rambow et al, 2015]) to follow the random migration of individual cells by time-lapse video microscopy followed by manual tracking of single cells (Fig 1D, see details in the Materials and Methods section). The MeWo cells are derived from lymph node metastasis of a cutaneous melanoma. They are tumorigenic and metastatic. They bear wild-type alleles at BRafV600 or RasQ61/G12 positions (Rambow et al, 2015; Ranzani et al, 2015) (Fig S2A). The A375M cells are derived from a human amelanotic melanoma. They are also tumorigenic and metastatic. They are mutated for BRafV600 and wild-type for RasQ61 (Rambow et al, 2015). Both cell lines actively migrated on Matrigel. Cells were tracked during 16 h in at least three independent experiments for each cell line (see Supplementary video microscopy). After PFKFB4 depletion using siRNA (Fig 1C), MeWo, and A375M cells migrated in average 33% and 42% slower than control, respectively (n = 27 independent biological replicates; Fig 1E and F and Table S1). Migration distance was also decreased while cell pausing was increased (Fig S2C–F). This de-fective migration after PFKFB4 depletion was also confirmed by a wound healing scratch assay in Mewo cells (Fig S2G and H). Moreover, to test if non-tumorigenic melanocytes also depended upon PFKFB4 for efficient migration, we used a cell line of spon-taneously immortalized wild-type melanocytes (12S2 cells) estab-lished from P4/P5 mice primary skin melanocytes (Valluet et al, 2012) and validated a siRNA against mouse PFKFB4 (Fig S7A and B). In this non-tumorigenic cell context, we also observe decreased cell migration (total distance) due to increased pausing time (Fig S7D–F). Because during *Xenopus laevis* embryonic development, the migration of melanocytes and melanoma progenitors, the neural crest cells, is also controlled by PFKFB4 (35), we postulated that human and frog protein functions were conserved, allowing us to devise phenotype rescue experiments: Frog *pfkfb4* encodes a protein with 95% similarity with the human protein, but the mRNA was not targeted by siRNAs designed against the human mRNA sequence. The frog PFKFB4 expression plasmid was efficiently translated in human melanoma cells and recognized by the an-tibody designed against human PFKFB4 (Fig 1C). In MeWo cells, the migration phenotype was efficiently rescued by co-transfecting the *X. laevis* orthologous *pfkfb4* sequence (Figs 1E and F and S2C). This rescue showed that the migration phenotype was specific for the depletion of PFKFB4 protein and did not come from off-target effects.

The best-known function for the bi-functional enzymes PFKFB1-4 is to regulate glycolysis rate by controlling the second rate-limiting reaction of glycolysis. We investigated if PFKFB4's role in cell mi-gration was linked with its function as a major activator of the glycolysis rate. To block glycolysis, we first grew the MeWo cells in a culture medium without glucose, which strongly decreased their glycolysis rate (estimated by the diminished lactate production measured in the culture medium, Fig S3B). We did not observe a major decrease of MeWo cells' average migration speed in the glucose-free medium compared with the complete medium con-dition (Fig 1G and H). Next, to confirm that MeWo cell migration was unaffected by limiting glycolysis, we added the glucose non-

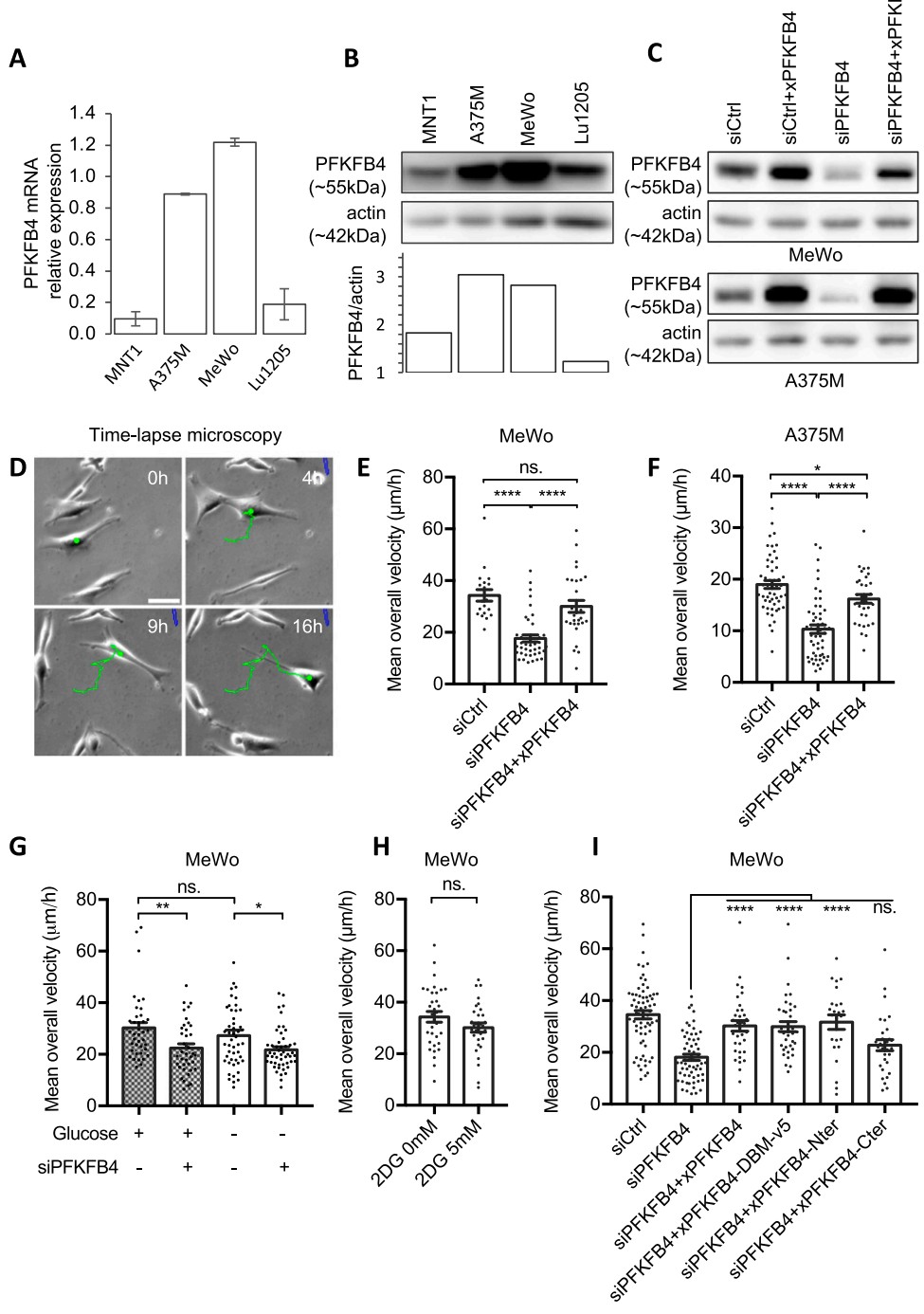

**Figure 1. PFKFB4 controls in vitro cell migration in metastatic melanoma in a glycolysis-independent manner.**
**(A, B)** Quantification of PFKFB4 mRNA and protein levels in four human melanoma cell lines: MNT1, A375M, MeWo, and Lu1205. **(A)** Relative *PFKFB4* mRNA levels measured by RT-qPCR and normalized by the expression of 18S and TBP housekeeping genes. Error bars: SEM. **(B)** PFKFB4 protein levels detected by Western-blotting, with PFKFB4/actin relative quantification (optical density). **(C)** PFKFB4 protein levels in MeWo and A375M cells 48 h after transfection with siRNA targeting PFKFB4 with or without co-transfection with *Xenopus laevis* PFKFB4 plasmid. **(D)** Starting 48 h after transfection, cell migration was tracked for 16 h from phase-contrast images. Scale bar is 20 μm. Each point corresponds to the average speed of one cell. **(E, F, G, H, I)** MeWo (E, G, H, I) or A375M (F) cells were co-transfected either with siControl/empty plasmid, siPFKFB4/empty plasmid, or with siPFKFB4 together with a *X. laevis* PFKFB4 plasmid in its wild-type form (E, F, I) or mutant forms (I). 27 independent biological replicates were performed, with 50–100 cells counted in each condition. Velocity reduction was in average of 33% for MeWo cells and of 42% for A375M cells. **(G, H)** The average speed was also measured when cells were cultured in glucose-free medium (G) or complete medium supplemented with 2DG (H). In each panel, a representative experiment is shown (n > 3), and displays mean ± SEM. *P*-values were calculated using the Mann–Whitney test. n.s.: $P > 0.05$; *$P < 0.05$; **$P < 0.01$; ***$P < 0.001$; ****$P < 0.0001$.
Source data are available for this figure.

hydrolysable analog 2-deoxyglucose (2DG) in the complete medium (Fig S3A). Similar to the glucose-free condition, 2DG also decreased glycolysis efficiently (Fig S3C) without affecting MeWo cells' migration speed (5 mM 2DG, Fig 1H). These results indicated that MeWo cells' migration was not directly linked to their rate of glycolysis. In contrast, PFKFB4-depleted MeWo cells showed a reduced average migration speed compared with control cells in the glucose-free medium as observed in the complete medium, indicating an action

of PFKFB4 on another cellular pathway (Fig 1G). Last, to test if PFKFB4 depletion was altering the overall rate of glycolysis, we performed two complementary approaches: first, we measured lactate production of MeWo cells after PFKFB4 depletion (Fig S3D); second we used the Seahorse XF Analyser to measure real-time extracellular acidification rate (ECAR) and oxygen consumption rate after successive addition of different metabolic inhibitors (first inhibitors of mitochondrial functions rotenone and antimycin A, followed by the

glycolytic inhibitor 2DG) using both MeWo and A375M cell lines (Fig S3E). From these measurements, we were able to assess different parameters of glycolysis such as the total proton efflux rate (PER) and the PER associated to glycolysis only (glycoPER) (Fig S3F and G) from which we evaluated the basal glycolysis and the compensatory glycolysis (Fig S3J and K). We did not observe any significant variation in lactate production (Fig S3D), in PER or glycoPER (Fig S3F–K), nor in compensatory glycolysis (Fig S3J and K). This is in agreement with previous reports indicating that PFKFB4 effect on glycolysis varies according to the cell context (Sakata et al, 1991; Okar et al, 2001; Ros et al, 2012; Ros & Schulze, 2013; Chesney et al, 2014). The characteristics of melanoma cell energy metabolism ensuring their migration in glucose-free conditions remain to be defined. Together, these observations suggested that PFKFB4 levels significantly affect the average speed of melanoma cells migration, independently of the rate of glycolysis.

PFKFB1-4 bi-functional enzymes possess two adjacent large catalytic regions. With their kinase moiety, PFKFB isoenzymes phosphorylate fructose-6-phosphate into fructose-2,6-bisphosphate (Pilkis et al, 1995; Okar et al, 2001) (Fig S3A). With their phosphatase domain, PFKFBs catalyze the reverse reaction. The two catalytic domains are highly conserved: the amino-acid sequence as well as the 3-dimensional protein structure are conserved both between isoenzymes in a given species and between species (Kotowski et al, 2021). To understand if PFKFB4 controlled melanoma cells migration using either its kinase or its phosphatase enzymatic activities, we compared the rescue phenotype of PFKFB4 depletion by various X. laevis PFKFB4 mutants (Fig S2B). The average speed of cells co-transfected with siPFKFB4 and a plasmid encoding a pfkfb4 mutant with two point mutations targeting the kinase and phosphatase enzymatic activities simultaneously (mutations G48A and H258A, [Tauler et al, 1990; Li et al, 1992]) was equivalent to the speed after rescue by wild-type PFKFB4 (xPFKFB4) (Fig 1I). This indicated that PFKFB4-controlled cell migration independently of its enzymatic activities. To identify which region of PFKFB4 protein was involved in this nonconventional effect, we used two complementary deletion constructs. The rescue done with a deletion construct encoding the N-terminal kinase domain (xPFKFB4-Nter) was as efficient as with xPFKFB4 (Fig 1I). In contrast, the migration of cells depleted for PFKFB4 and co-transfected with the deletion construct encoding only the C-terminal phosphatase domain (xPFKFB4-Cter) was not significantly rescued (Fig 1I). This result suggested that PFKFB4 was involved in control of cell migration independently of its kinase or phosphatase activities, but through the N-terminal half of the protein. We next checked if PFKFB4 depletion altered cell cycle or cell death, using FACS analysis. We did not detect variation in cell apoptosis nor in the relative duration of cell cycle phases either in MeWo or in A375M cells (Fig S4). Altogether these results suggest that PFKFB4 regulated the efficiency of melanoma cell migration independently of variations in glycolysis, cell survival rate or cell cycle.

### PFKFB4 interacts with ICMT, a major posttranslational modifier of RAS GTPases

As PFKFB4 protein seemed to control melanoma cell migration independently of its enzymatic activities, we looked for interacting protein partners using mass spectrometry after immunoprecipitation of a FLAG-tagged form of human PFKFB4 expressed in the MeWo cells. In two biological replicates, among 1,556 high confidence hits, we chose 40 candidates with a Mascot score enriched at least ten-times compared with the negative control condition to eliminate the weak hits and limit the nonspecific targets. Moreover, because xPFKFB4 efficiently rescued the PFKFB4 depletion phenotype in human melanoma cells, we postulated that the protein function we looked for was evolutionarily conserved between human and X. laevis PFKFB4. We transfected MeWo cells with the frog xPFKFB4 ortholog followed by immunoprecipitation and mass spectrometry. We then crossed the 40 candidates list obtained with hPFKFB4 with the list of xPFKFB4 targets and sub-selected 22 candidates (Figs 2A and S5). Among these 22 best candidates, we prioritized isoprenylcystein carboxyl methyl transferase (ICMT), a potential modulator of PI3K/AKT signaling pathway, because PFKFB4 was known to affect cell migration via AKT signaling activation during embryogenesis (Pegoraro et al, 2015; Figueiredo et al, 2017). ICMT is an endoplasmic reticulum membrane protein critical for RAS GTPases posttranslational modifications. ICMT catalyzes the carboxyl methylation of RAS on its C-terminal CAAX motif. This modification allows RAS protein to be targeted to the plasma membrane, a prerequisite for the coordination by RAS of a variety of signaling pathways, including PI3K/AKT activation ([Dai et al, 1998; Choy et al, 1999; Michaelson et al, 2005; Wright et al, 2009], reviewed in Cansado [2018]). This observation suggested that an interaction between PFKFB4 and ICMT could occur during melanomagenesis and be related to RAS-dependent signaling pathways.

First, we validated the mass spectrometry results by co-immunoprecipitation of ICMT with PFKFB4 in MeWo cells (Fig 2B). Second, we confirmed the protein–protein interaction between PFKFB4 and ICMT by an independent alternative method, a split-ubiquitin two-hybrid approach adapted for mammalian membrane proteins (MaMTH) (Fig 2C) (Petschnigg et al, 2014; Saraon et al, 2017). Briefly, ICMT and PFKFB4 were fused to a portion of the ubiquitin protein, either its N-ter part (Nub, N-ubiquitin) or its C-ter part fused to GAL4 (Cub-GAL4, C-ubiquitin). Constructs were co-transfected in MaMTH-modified HEK293T cells bearing a stable integration of GAL4-binding sites upstream of a luciferase reporter. Upon interaction, the two halves of ubiquitin reunite to form a "pseudo-ubiquitin" which recruits deubiquitinating enzymes (DUBs). The DUBs then cleave the pseudo-ubiquitin, resulting in the release of the GAL4 transcription factor. GAL4 then activates the transcription of GAL4-driven luciferase in the nucleus. As a positive control, co-transfection of PFKFB4-Cub-GAL4 and PFKFB4-Nub strongly increased luciferase expression (by five to thirteen times compared with PFKFB4-Cub-GAL4 alone, for PFKFB4-Nub fusion in N-ter or C-ter, respectively). This denoted a strong and stable interaction, related to the formation of the PFKFB4 homo-dimer. The co-transfection of ICMT-Cub-GAL4 and PFKFB4-Nub significantly increased luciferase expression compared with ICMT-Cub-GAL4 alone and in a range comparable with the known PFKFB4-PFKFB4 homophilic interaction. Together, these results demonstrated that PFKFB4 and ICMT directly interacted. Last, we tested if the interaction between PFKFB4 and ICMT was important for the known interaction between ICMT and RAS GTPase. When PFKFB4 was depleted in MeWo cells, we observed a decrease of

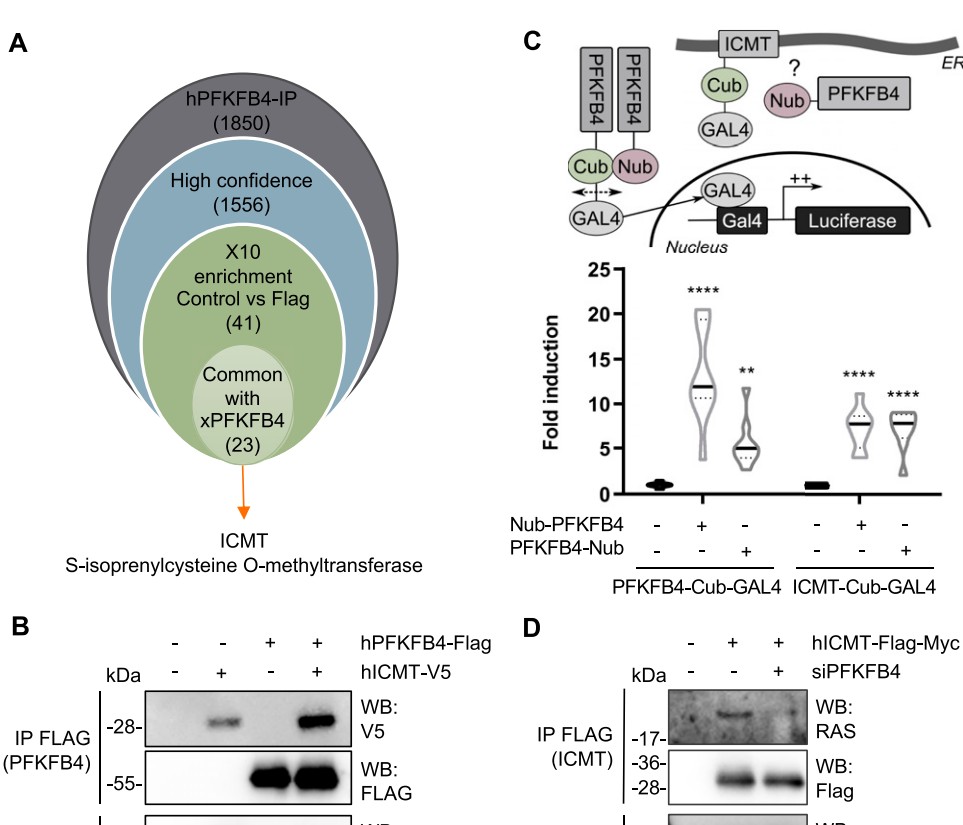

**Figure 2. PFKFB4 interacts with ICMT, a major posttranslational modifier of RAS GTPases.**
**(A)** Workflow used to select candidates after PFKFB4 immunoprecipitation followed by mass spectrometry analysis (see text for details). **(B)** Enrichment of ICMT tagged with V5 after immunoprecipitation by FLAG PFKFB4 from MeWo cell extracts. **(C)** Scheme of the MaMTH strategy used to validate PFKFB4/ICMT protein–protein interactions. Violin plot showing the luciferase activity measured and normalized from MaMTH-modified HEK293T cells extracts (n = 3). The violin represents the probability density at each value; lines are plotted at the median and quartiles (Two-way ANOVA test. **P < 0.01 and ****P < 0.0001). **(D)** The interaction between tagged ICMT and endogenous RAS was evaluated with or without PFKFB4 depletion (n = 2). Immunoprecipitation of FLAG-ICMT from MeWo cells followed by Western blotting with antibody against V5, FLAG, or endogenous RAS.
Source data are available for this figure.

endogenous RAS immunoprecipitation by ICMT (Fig 2D). In sum, all these results suggested that PFKFB4 direct protein–protein interactions with ICMT impacted ICMT–RAS complex formation in melanoma.

### ICMT and PFKFB4 control RAS localization at the plasma membrane and melanoma cell migration

To understand the role of the PFKFB4-ICMT interaction, we first compared PFKFB4 and ICMT depletion phenotypes in MeWo cells, using a validated siRNA against ICMT (Cansado, 2018) (Fig S6A). Parameters of melanoma cell migration were measured as mentioned previously. Compared with control siRNA, cells transfected with siICMT exhibited a decrease in their average speed of migration, as well as altered pausing and distance parameters, similar to cells transfected with siPFKFB4 (Figs 3A and S6B and C). To test the interdependency of PFKFB4 and ICMT, we co-transfected both siRNAs. Melanoma MeWo cells receiving both siPFKFB4 and siICMT did not exhibit a more severe phenotype than with either siRNA alone. This suggested that PFKFB4 and ICMT cooperated in the same pathway to control cell migration and that depleting either one was sufficient for attaining a strong phenotype (Fig 3A). To further test this hypothesis, we tested the epistasis between PFKFB4 and ICMT by combining depletion of one factor and gain-of-function of the other, to see if increased activity of either one of these proteins

could compensate for the loss of the other, as could be the case if they were acting in parallel and redundant pathways: the siPFKFB4 was co-transfected with the *ICMT* expression plasmid, or the siICMT with the *PFKFB4* expression plasmid. When compared with the migration speed of MeWo cells transfected with a control siRNA and the corresponding siRNA alone, neither ICMT nor PFKFB4 gain-of-function rescued the phenotype of PFKFB4 or ICMT depletion, respectively (Fig 3B). This suggested two alternative (and not exclusive) possibilities: either the need for both proteins simultaneously, cooperating to enhance cell migration, or that one of these two proteins was functional only after being activated by the other. As a whole, this series of results showed that ICMT depletion phenocopied loss of PFKFB4, and that the two protein partners were likely acting in the same pathway impacting melanoma cell migration.

In parallel, we compared PFKFB4 depletion phenotype with the known effect of ICMT depletion, as the major role of ICMT is to modify RAS proteins posttranslationally for their efficient targeting to the plasma membrane (Michaelson et al, 2005). To be free from potential defects in RAS GTPase activity, we have used a constitutively active form of RAS, HA-tagged-RasV12, and tested if PFKFB4 influenced its subcellular localization, by immunofluorescence. After co-transfecting MeWo cells with HA-tagged-RasV12 and either siPFKFB4, siICMT, or a control siRNA, we scored RAS localization to the plasma membrane qualitatively (Fig 3D). On top of the

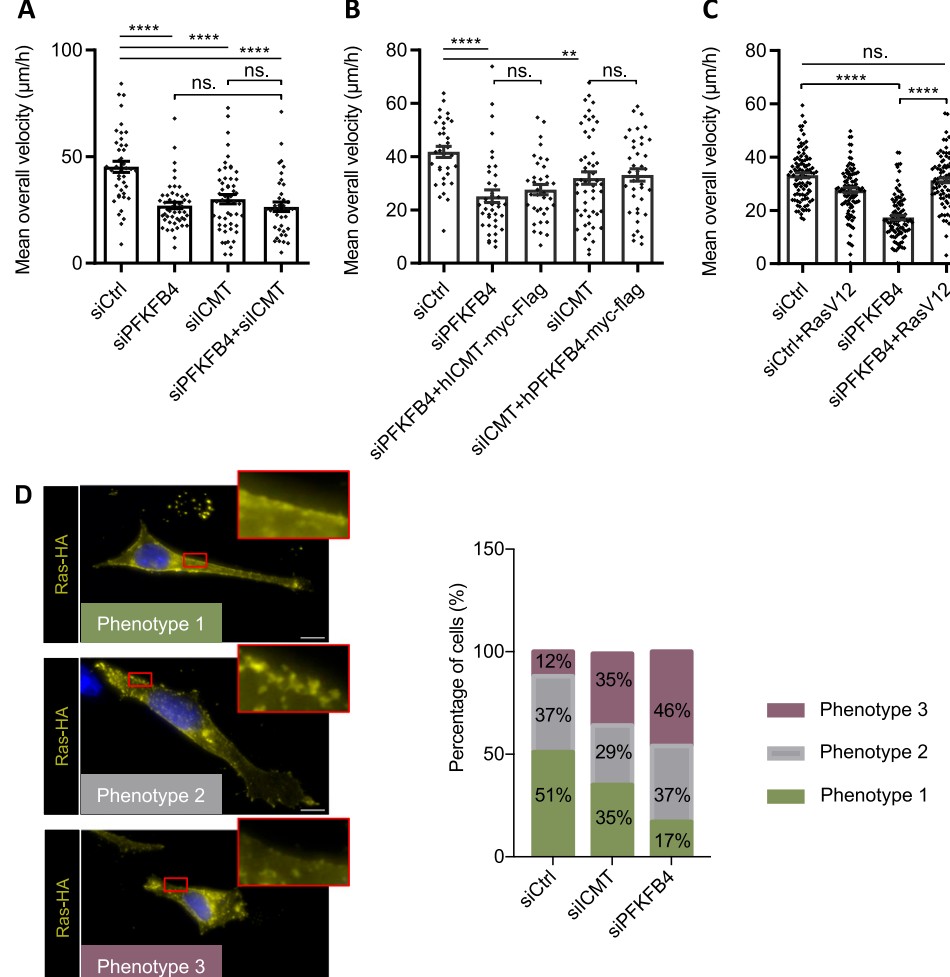

**Figure 3.  ICMT and PFKFB4 both control RAS addressing at the plasma membrane and melanoma cell migration.**
**(A, B, C)** Average speed of MeWo cells transfected with siRNA (siCtrl, siPFKFB4, or siICMT) and plasmid (empty vector, hICMT-myc-Flag, hPFKFB4-myc-Flag, or RasV12). **(A, B, C)** Graphs show the mean calculated in one experiment with at least 30 cells in each condition (A: n = 1, B: n = 1, C: n = 3), Error bars are calculated with SEM. *P*-values are calculated using the Mann–Whitney test. n.s.: $P > 0.05$; *$P < 0.05$; **$P < 0.01$; ***$P < 0.001$; and ****$P < 0.0001$. **(D)** Detection of RAS subcellular localization by immunostaining on MeWo cell transfected with RasV12-HA (yellow). Nuclei were stained with DAPI (blue). At the plasma membrane, RAS was found distributed according to three main phenotypes: either a clear and homogeneous membrane localization (Phenotype 1), or the absence of signal (Phenotype 3), or an intermediate phenotype with intermittent RAS expression at the membrane (Phenotype 2). Insets show enlargements of the areas framed in red. Scale bar is 10 $\mu$m. The proportion of each phenotypes was quantified after transfection of either siControl (nbcell = 43), or siICMT (nbcell = 51), or siPFKFB4 (nbcell = 46). A representative experiment is shown, n = 3. Source data are available for this figure.

exogeneous RasV12 perinuclear location, the cells transfected with the control siRNA could be categorized into three groups: cells exhibiting a clear and homogeneous RAS membrane enrichment localization (phenotype 1), cells without RAS membrane enrichment (phenotype 3), and cells with an in-between phenotype with discontinuous RAS membrane localization (phenotype 2) (Fig 3D). Whereas phenotype 2 was present in a similar proportion in each condition (siControl, siICMT, or siPFKFB4; 30–37% of the cells), cells with RAS at the plasma membrane represented 51% of the siControl cells, but only 35% of siICMT cells and 17% of siPFKFB4 cells (Fig 3D). The remaining cells displayed phenotype 3. This assay thus showed that PFKFB4 depletion phenocopied ICMT loss for RAS addressing to the plasma membrane in melanoma cells.

Last, to test if RAS was indeed a downstream target of PFKFB4 in the control of cell migration, we measured MeWo and A375M cell migration efficiency after co-transfecting siPFKFB4 with the constitutively active RasV12. In both cell lines, RasV12 rescued the PFKFB4 migration phenotype (Figs 3C and S6D). We noticed that in this case, RasV12 discontinuous membrane localization (Fig 3D) was sufficient to restore cell migration functionally (Fig 3C), probably because of the constitutive and enhanced activity of the RasV12

protein. From this series of results, we concluded that the interaction between PFKFB4 and ICMT was critical for controlling RAS subcellular localization and melanoma cell migration. PFKFB4 thus displays a novel function, important to modulate the activity of a major cell signaling pathway in cancer cells.

## PFKFB4 and ICMT control RAS-AKT signaling in melanocytes and in melanoma

RAS signaling is a major hub for the cell to integrate multiple inputs from the extracellular cues as well as from intracellular parameters. Once activated, several downstream pathways are activated in normal as well as cancer cells. The major ones include MAP kinase, PI3 kinase, and Ral signaling cascades (Rodriguez-Viciana et al, 1994; Urano et al, 1996; Peyssonnaux et al, 2000). In melanoma, MAPK/Erk and PI3K/AKT pathways are frequently activated to control cell migration. Here, we next sought to further understand which of these two pathways was modulated by PFKFB4 depletion (Fig 4). Whereas ERK phosphorylation (pERK) remained unchanged after transfection of MeWo cells with siPFKFB4 or siICMT, AKT phosphorylation was strongly decreased both on threonine 308

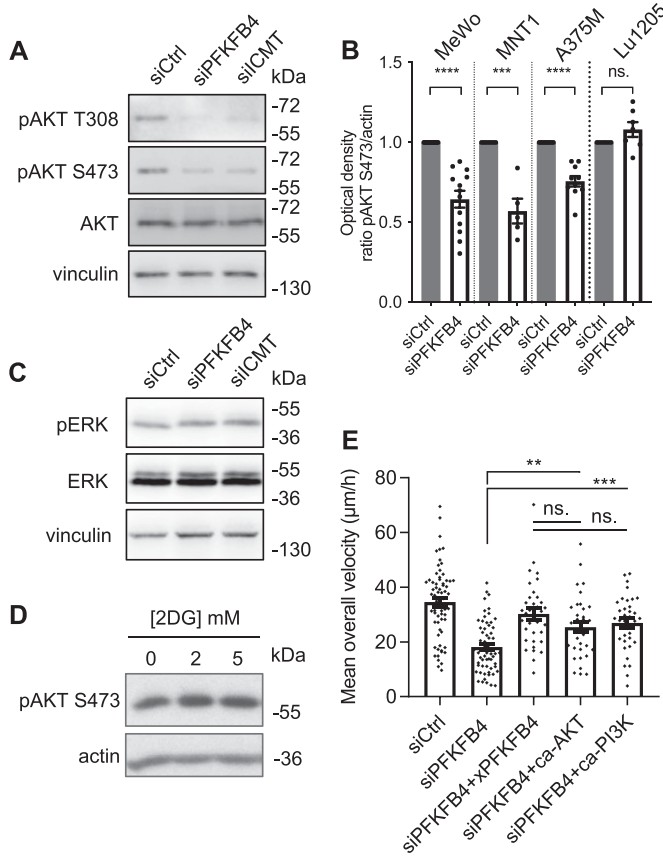

**Figure 4. PFKFB4 and ICMT both control AKT signaling activation in melanoma cells.**
**(A)** Protein levels of pAKT T308, pAKT S473, AKT, and actin in MeWo cells transfected with siRNA targeting PFKFB4 or ICMT. **(B)** Normalized pAKT S473/actin levels in MeWo, MNT1, A375, or Lu1205 cells transfected with siRNA targeting PFKFB4 and analyzed as in A; one point represents one biological replicate. **(C)** Protein levels of pERK, ERK, and vinculin in MeWo cells transfected with siRNA targeting PFKFB4. **(D)** Protein levels of pAKT S473 in MeWo cells treated with different concentrations of 2DG for 24 h. **(E)** Average speed of MeWo cells transfected either with siControl+empty vector, siPFKFB4+empty vector, siPFKFB4+xenopus PFKFB4 wild-type, siPFKFB4+caAKT, and siPFKFB4+ca-PI3K. In (A, C, D, E): a representative experiment is shown, n > 3. Graphs show the mean ± SEM. *P*-values were calculated using the Mann–Whitney test. n.s. *P* > 0.05, \*\**P* < 0.01, \*\*\**P* < 0.001, and \*\*\*\**P* < 0.0001.
Source data are available for this figure.

and on serine 473, the two major modifications leading to full activation of AKT (Figs 4A–C and S6E). To extend this finding to other melanoma contexts, we examined AKT activation in three other cell lines after PFKFB4 depletion. In all cell lines, siRNA against PFKFB4 reduced target protein levels efficiently (Fig S6F and G). Phospho-AKT levels were significantly decreased in A375M and MNT1 cells, whereas there was no significant decrease in Lu1205 (Fig 4B). This suggested that PFKFB4 influenced AKT signaling in several but not all melanoma cell contexts. Although it remains unclear why MAP kinase signaling remains unaffected, whereas AKT pathway is decreased in this particular context, similar examples of selective activation of the PI3K-AKT pathway have been described before (Posch et al, 2016). Last, we found that AKT phosphorylation was also significantly decreased after PFKFB4 depletion in primary mouse 12S2 melanocytes (Fig S7C).

We then wondered if PFKFB4 might affect AKT signaling as an indirect effect of glycolysis regulation. We blocked glycolysis with 2DG and tested AKT activation in MeWo cells. Although decreased lactate levels indicated an efficient block of glycolysis (Fig S3), the treatment with 2DG did not affect AKT phosphorylation on S473 (Fig 4D). As observed above for the cell migration phenotype (Fig 1), AKT phosphorylation phenotype after depleting PFKFB4 was thus not likely due to a reduction of glycolysis. Last, we expressed constitutively active forms of either AKT (caAKT) or its upstream regulator PI3 kinase (caPI3K) in PFKFB4-depleted MeWo cells. Defective cell migration parameters observed after PFKFB4 depletion were rescued either by caAKT or by caPI3K (Figs 4E and S6H and I). All these data together indicated that PFKFB4 controlled human melanoma cell migration via a novel nonconventional function controlling the RAS/PI3K/AKT pathway.

## Discussion

PFKFB proteins are long-known major regulators of the rate of glycolysis in normal and cancer cell types (Rider et al, 2004). They have been involved in mediating the Warburg effect in many different tumors, and have been shown to be activated in response to hypoxic conditions often found at the heart of primary tumors (Minchenko et al, 2005a, 2005b, 2014; Yun et al, 2012; Chesney et al, 2014; Zhang et al, 2016a). In particular, elevated levels of *PFKFB4* expression have been described in melanoma (Fig S1, [Minchenko et al, 2005b]). Using a survey of human melanoma cell lines transcriptomes (Rambow et al, 2015), we have selected cells with high *PFKFB4* levels, and explored PFKFB4 function in the biology of those cells, focusing on their migration in vitro. We first showed that PFKFB4 enhanced cell migration irrespective of the cells' glycolysis levels (Figs 1 and S2). Moreover, neither PFKFB4 kinase nor its phosphatase activity was required for this effect, suggesting alternative molecular mechanisms, such as protein–protein interactions. Using PFKFB4 immunoprecipitation followed by mass spectrometry, we have identified a partner of PFKFB4 which was selected for further analysis: ICMT, an enzyme embedded into the endoplasmic reticulum membrane, which adds the terminal methyl group to RAS GTPases posttranslationally. This modification is required for anchorage of RAS GTPases at the plasma membrane, where RAS activates several downstream signaling events (Cox & Der, 2010). Strikingly, both PFKFB4 and ICMT depletion displayed similar phenotypes including RAS mislocalization, decreased AKT activation and reduced cell migration (Figs 2–4). Migration phenotypes were rescued by a constitutively active form of RAS or by the constitutive activation of AKT signaling (Fig 5). In sum, our study has demonstrated a novel, glycolysis-independent function of PFKFB4, promoting the interaction between ICMT and RAS, resulting in active migration of both melanoma cells and melanocytes (Figs 5 and S7).

Our main thought-provoking finding is that the regulation of cell migration by PFKFB4 does not depend on its kinase activity and that PFKFB4 depletion affects migration independently of experimental modulations of the high/low glycolysis status of the cells (Figs 1 and S2). Our results thus uncouple the classical action of PFKFB4 in glycolysis from its role in cell migration. This undermines current

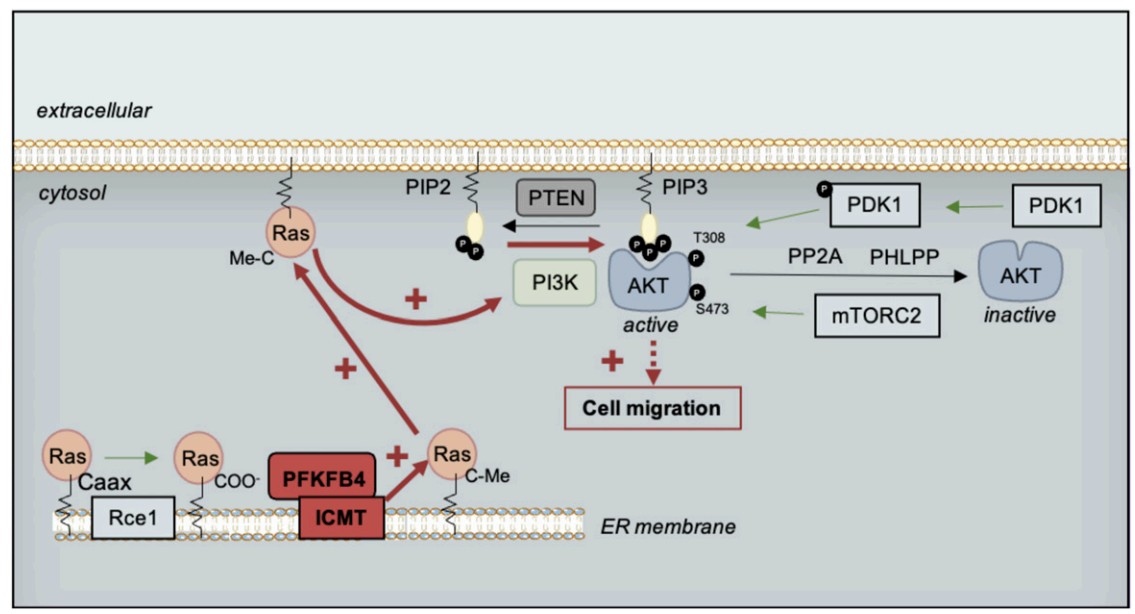

**Figure 5.  Model of cell migration control by a noncanonical function of PFKFB4, modulating RAS signaling.**
We propose that the interaction between PFKFB4 and ICMT would promote the ICMT/RAS interaction needed for RAS trafficking to the plasma membrane, where RAS would activate PI3K-mediated AKT phosphorylation on T308. In turn, AKT activation would then modulate cell migration. PTEN, Phosphatase and Tensin homolog; PI3K, phosphoinositide 3-kinase; PIP2, phosphatidylinositol-4,5-bisphosphate; PDK1, Phosphoinositide-dependent kinase 1; PP2A, Protein phosphatase 2; PHLPP, PH domain and Leucine rich repeat Protein Phosphatase; mTORC2, mTOR complex 2; Rce1, Ras converting enzyme 1; ICMT, isoprenylcysteine carboxyl O-methyl transferase.

strategies to counteract PFKFB4 in cancer, which involves developing pharmacological drugs to interfere with PFKFB4 kinase activity, assuming that PFKFB's major function in cancer is to promote glycolysis and Warburg effect (Chesney et al, 2015). Our data show that a kinase-deficient form of PFKFB4 still retains important cancer-promoting functions outside of glycolysis regulation. Further biochemical and crystallographic analyses, beyond the scope of this cell biology study, would provide details on the protein–protein interacting subdomains interfacing PFKFB4 and ICMT. The disruption of this interaction could be an additional strategy to block PFKFB4 in cancer.

ICMT is a major posttranslational modifier of RAS GTPases. ICMT catalyzes RAS terminal methylation on the ER membrane, needed to address RAS to the plasma membrane (Dai et al, 1998; Choy et al, 1999; Michaelson et al, 2005). One-third of human cutaneous melanoma are mutated on RAS, PI3 kinase, or other partners of this pathway, enhancing its signaling activity (Ali et al, 2013; Cancer Genome Atlas Network, 2015). However, here we showed that PFKFB4 promoted RAS signaling and cell migration in two cell lines with wild-type NRAS. This implied that, even in the absence of a RAS activating mutation, an increase in PFKFB4 cellular levels might also allow enhanced RAS-linked signaling and cell migration. Increase in *PFKFB4* gene expression can be achieved by hypoxia, a general feature of growing tumors. HIF1α-responsive elements have been identified to promote *pfkfb4* transcription (Minchenko et al, 2004). Whereas in normal cells, there is a fine-tuned, dynamic, and tissue-specific expression of PFKFB genes during development and cell homeostasis (Minchenko et al, 2003; Pegoraro et al, 2013, 2015), it is likely that tumor progression enables a hypoxia-induced broader and sustained expression of PFKFB4, which would in turn promote tumor cell migration in parallel to its activation of Warburg effect.

However, in melanoma cell lines, a tumor type with generally high *PFKFB4* levels, we did not observe a strict correlation between PFKFB4 expression levels and the metastatic characteristics of the cells (Fig S2). This indicates the importance of yet unknown additional cell-specific cues.

We have focused on the function of RAS-AKT signaling in the control of melanoma cell migration. This study makes a further parallel between melanoma cell features and the behavior of their parent cells in embryos, the neural crest cells. PFKFB4 was first identified as a regulator of cell migration in neural crest, and as a general patterning regulator during neural and neural crest early development (Pegoraro et al, 2015; Figueiredo et al, 2017). Melanoma initiation and progression involves the reactivation of elements belonging to the neural crest developmental program (Kaufman et al, 2016). We here extend the parallel between the two models, showing increased expression of PFKFB4 in development and cancer as predicted by our previous WGCNA analyses (Plouhinec et al, 2017). Moreover, in the embryonic cells, which rely on yolk breakdown for their energy metabolism rather than on glycolysis, we have revealed the first indications for a nonconventional function of PFKFB4. This function involved enhancing AKT signaling and cell migration (Pegoraro et al, 2015; Figueiredo et al, 2017). It remained unclear whether this novel function of PFKFB4, found in a nonmammalian in vivo model, was also important in mammalian cells. Here, human melanoma cells present similar regulations by PFKFB4, implicating AKT signaling and the control of cell migration. This study thus further emphasizes the importance of PFKFB4 "moonlight" or nonconventional signaling functions, a term naming a function which is revealed when the major "sunlight" function is masked (here, the key control of glycolysis rate by PFKFB proteins).

Although we stress the importance of the PFKFB4-ICMT-RAS-AKT signaling pathway, we do not exclude parallel important functions for other signaling proteins that modulate cell migration: our mass spectrometry screen for PFKFB4 partners has provided about 20 other strong interaction candidates, which could also be involved in cell trafficking or cell migration (Figs 2 and S3). In conclusion, our study highlights a novel and unsuspected link between three major hallmarks of cancer cells, namely, cell metabolism, signaling, and migration. The crosstalk between key regulators of glycolysis and Warburg effect (PFKFB4) and a pleiotropic cell signaling pathway (ICMT–RAS) further increases the complexity of the network known to promote melanoma cancer progression. Those intricate relationships, which might also act as re-wiring options upon cancer treatment and relapse, will be important targets for future therapeutic options.

# Materials and Methods

### Cloning, plasmids

All plasmids used are listed in Table S2. For testing protein–protein interactions, a two-hybrid-like assay adapted for mammalian membrane-bound proteins (MaMTH) was used (Petschnigg et al, 2014; Saraon et al, 2017). Cloning used Gibson method ([Gibson et al, 2009], primers used in Table S3). hICMT (clone Origene no. RC207000) and hPFKFB4 (clone Origene no. RC201573) were inserted in-frame into the MaMTH bait destination vector, which contains ubiquitin C-terminal half fused to the yeast GAL4 DNA-binding domain, or into the C-tagged or N-tagged MaMTH prey destination backbone vector, which contains ubiquitin N-terminal half.

### Cell lines, cell culture, cell treatments, and cell transfection

The well-characterized human melanoma cell lines MeWo (Kerbel et al, 1984; Ishikawa et al, 1988), A375M (Sriramarao & Bourdon, 1996), MNT1 (Cuomo et al, 1991), Lu1205 (Juhasz et al, 1993) were kindly provided by Dr. L Larue (Rambow et al, 2015). Their mutagenic status for key driver mutations in melanoma is summarized in Fig S2 (Rambow et al, 2015; Ranzani et al, 2015). Cells were cultured in RPMI (Gibco) supplemented with 10% SVF and 1% penicillin/streptomycin (Invitrogen). HEK293T cells were cultured in DMEM (Gibco) supplemented with 10% SVF and 1% penicillin/streptomycin (Invitrogen). The wild-type mouse melanocytes 12S2 cells were kindly provided by Dr S Druillennec. They were established from the skin of wild-type mouse with a mixed background (C57Bl/6&129/SV) which are wild type for the three Ras isoforms (Valluet et al, 2012). 12S2 mouse melanocytes were grown as in Valluet et al (2012) in F12 Nutrient Mixture (#21765-029; Gibco) supplemented with 10% FBS, 1% penicillin/streptomycin (P/S), and 200 nM phorbol-12-myristate-13-acetate (TPA) (P8139; Sigma-Aldrich). All cell lines were incubated at 37°C with 5% $CO_2$. At 24 h before transfection, cell lines were plated at 200,000 cells per well (A375M and HEK293T) or 300,000 cells per well (MNT1, MeWo, Lu1205, 12S2) into six-well plates. For siRNA experiments, human melanoma cells were transfected either with a control siRNA (Stealth Negative Control

Medium GC Duplex, Invitrogen) or with siPFKFB4 (Dharmacon Smartpool siGenome D-006764-01/02/04/17, siPFKFB4 [1]; Invitrogen #HSS107863, siPFKFB4 [2]), or with siICMT (Dharmacon Smartpool siGenome #M-005209-01-0010) (Table S1) at 30–90 pM using lipofectamine RNAimax (Invitrogen) according to the manufacturer's instructions. Mouse 12S2 primary melanocytes were transfected either with a control siRNA (Stealth Negative Control Medium GC Duplex, Invitrogen) or with siPFKFB4 against mouse sequences (Dharmacon Smartpool M-054640-01-0010). For the gain-of-function experiments, cells were transfected with a total of 0.5–1 µg of DNA using lipofectamine 2000 (Invitrogen). For Check-Mate experiments, 0.5 µg of each plasmid (pBind/pG5 or pAct/pG5) were transfected using Lipofectamine 2000. For glycolysis blockade, a RPMI glucose-free medium was used (Gibco). Alternatively, 2–5 mM of 2-deoxy-glucose (Sigma-Aldrich) was added to the normal medium. All experiments were analyzed 48 h after transfection (Figs 1–4).

### Luciferase assay

A luciferase reporter driven by five GAL4-binding sites (pG5-luc) was co-transfected into HEK293T cells together with the plasmids to be tested and a control plasmid coding Renilla luciferase for normalization of the signals. Cell medium was changed 24 h after transfection. At 48 h, cells were rinsed with PBS and lysed with passive lysis buffer 1X (Promega) for 15 min with agitation at room temperature. Firefly and Renilla luciferase activities were measured using the Dual-Glo Luciferase Assay System (Promega). For each condition, signal intensity was normalized by the ratio between Firefly and Renilla luminescence. Transfections were performed in triplicate, each with technical duplicates.

### Protein extraction and Western-blotting

Cells were washed in PBS and lysed in RIPA buffer (10 mM Tris–HCL, pH 8, 150 mM NaCl, 1% NP-40, 0.1% SDS, and 0.5M sodium deoxycholate) supplemented with phosphatases inhibitor (Sigma-Aldrich) and proteases inhibitors (Sigma-Aldrich) at 4°C. Protein samples were resolved on 12% SDS–PAGE gels and transferred to PVDF membranes (Bio-Rad). After blocking in 5% skimmed milk diluted in TBS–0.1% Tween (TBS-T), membranes were probed with primary antibody diluted in the blocking buffer overnight at 4°C (dilutions are indicated in Table S4). After three washes in TBS-T, membranes were probed with HRP-conjugated goat anti-rabbit or anti-mouse (1:20,000) 1 h at room temperature. ECL signal was quantified by densitometric analyses using ImageJ software (http://rsb.info.nih.gov/ij/).

### Co-immunoprecipitation and mass spectrometry

At 48 h after transfection, total proteins were extracted using a mild lysis buffer (100 mM NaCl, 0.5% NP-40, 20 mM Tris–HCl, pH 7.5, 5 mM $MgCl_2$, protease inhibitors, and phosphatases inhibitors). Cells were lysed by mechanical passages through a 26-gauge syringe. For each sample, 20 µl of FLAG-M2 magnetic beads (M8823; Sigma-Aldrich) previously washed in lysis buffer were added and incubated with agitation overnight at 4°C. Beads were then washed five times in

lysis buffer. For mass spectrometry analysis, beads were further washed twice with $H_2O$. Proteins were digested with trypsin, desalted using ZipTip C18 and analyzed using a nanoESI-Orbitrap Fusion (Thermo Fisher Scientific). Data were analyzed using Mascot (Matrix Science). Mascot scores of the negative control, here tagged-V5 PFKFB4, were compared with the sample-tagged FLAG PFKFB4 protein, as advised by the platform. For analysis by Western blotting, the co-immunoprecipitated proteins were eluted by boiling 10 min in Laemmli buffer (50 mM Tris, pH 6.8, glycerol, 2% SDS, 3% DTT, and bromophenol blue), or eluted by competition with Flag peptide at 200 $\mu g/ml$ (five incubations of 5 min, P4799; Sigma-Aldrich). Samples were finally concentrated using 3 kD columns (Millipore).

### RNA extraction and RT-qPCR

Total RNA was extracted from cells lysed in Trizol (Thermo Fisher Scientific), and then purified by chloroform extraction and isopropanol precipitation. We used M-MLV reverse transcriptase (Promega) for reverse transcription and SYBR Green mix (Bio-Rad) for quantitative PCR. Results were normalized against reference genes *tbp* and *18S* (see Table S5 for sequences).

### L-lactate dosage

Cell supernatant was collected 24 h after changing cell medium then immediately filtered by centrifugation for 15 min at 4°C on a 10 kD column (Abcam or Millipore). This step eliminates proteins, including lactate dehydrogenase (LDH) to avoid nonspecific L-Lactate degradation in the sample. Extracellular L-lactate concentration was measured from the filtered medium using the L-Lactate kit (ab65330; Abcam). Normalization was done according to total protein concentration of the sample which is directly proportional to the total cell number.

### Seahorse experiments

Glycolysis was assessed by measuring oxygen consumption rate and extracellular acidification rate (ECAR) using a Seahorse XFe96 analyzer (Agilent Technologies) at the Biomarkers Platform at Pasteur institute (Paris), to determine the PER and PER associated to glycolysis only (glycoPER). Briefly, MeWo and A375M cell lines were transfected as describe above 48 h before the assay and were seeded on Seahorse XF96 cell culture microplate (Agilent Technologies) at a density of 10,000 cells/well 24 h before the assay. The day of the assay, the medium was replaced by Seahorse XF DMEM medium, pH 7.4 (Agilent Technologies), supplemented with 1 mM pyruvate (Agilent Technologies), 2 mM glutamine (Agilent Technologies), and 10 mM glucose (Agilent Technologies) and incubated 1 h in a non-$CO_2$ incubator. Just before the assay, cell medium was changed for fresh warm assay medium. A final well concentration of 0.5 $\mu M$ of rotenone and antimycin A mixture and 50 mM of 2-deoxy-D-glucose (2-DG) were sequentially injected to calculate PER and glycoPER. Data were analyzed using the Wave software 2.6.3 (Agilent Technologies) and normalized to cell confluency (in %) obtained by phase-contrast imaging and analyzed with Incucyte SX5 (Sartorius).

Curves were plotted using RStudio; superplots and statistical analysis were generated using SuperPlotsofData (Goedhart, 2021).

### Two-dimensional random cell migration assayed by time-lapse video microscopy

We dispensed 40,000 or 50,000 cells per well into twelve-well plates coated with Matrigel (Thermo Fisher Scientific) or 40 ng/$\mu l$ of fibronectin (#F1141; Sigma-Aldrich) (for 12S2 or MeWo/A375M, respectively). Two-dimensional (2D) random cell migration was monitored by time-lapse video microscopy under bright white light, with an inverted phase-contrast microscope (Leica MM AF) equipped with a cell culture chamber (37°C, humidified atmosphere containing 5% $CO_2$), an x–y–z stage controller, and a charge-coupled device CoolSnap camera (Photometrics). Images were acquired at 8-min intervals during 16 h, with the Metamorph software (Molecular Devices). Movies were reconstructed with the ImageJ software (http://rsbweb.nih.gov/ij/). Cells were tracked manually by using the center of the nucleus as guide and parameters were calculated with ImageJ Manual Tracking plug-in. Individually tracked cells were chosen to be alive by eye monitoring over the entire duration of the movie. Briefly, the manual tracking plug-in is recording x and y positions of each cells tracked and then generate statistical values as the mean overall velocity (in $\mu m/h$), total distance travelled (in $\mu m$) or % pausing (which is the cumulative fraction of time of the total duration of the movie where the cells are not changing position).

### Wound healing/scratch assay

We dispensed 60,000 MeWo cells per well in a 96-well plate and incubated it in a humidified incubator at 37°C and 5% $CO_2$. When the cells reached confluency, a wound field was made using Incucyte WoundMaker (Essen Bioscience) followed by three washes to remove debris.

The healing was monitored every 3 h for 90 h by Incucyte Live-Cell Imaging Systems (Essen Bioscience) with a ×10 objective. The relative cell density at the wound was calculated by Incucyte analyzer after training on a subset of images, providing real-time cellular confluence data, based on segmentation of high-definition phase-contrast images. The relative wound cell density is defined as the percent of confluence in the wound area compared with the confluency outside of this region. Raw data were then exported to Prism software for plotting and statistical analysis using ANOVA test.

### FACS sorting, cell cycle, and cell death analysis

Standard protocols were used. Briefly, for cell death analysis, cells were trypsinized and resuspended in PBS containing 0.25 $\mu g/ml$ of 7-aminoactinomycin D (7-AAD). Cell suspension was analyzed on FACSCANTO II cell sorter. Percent of stained (dead) cells was calculated. For cell cycle analysis, cells were trypsinized, rinsed in PBS, fixed in 70% ethanol, and stored at −20°C overnight. After several PBS washes, cells were resuspended in 3.5 mM Tris HCl, pH 7.6, 10 mM NaCl, 1 $\mu g/ml$ 7-AAD, 0.1% NP-40, 40 U/$\mu l$ RNAse A, and incubated in the dark for 30 min. The cell suspension was sorted on

FACSCANTO II cell sorter. The relative importance of each phase of the cell cycle was automatically calculated using standard modeling algorithms of the FlowJo software.

### Immunofluorescence

MeWo cells were plated 24 h before transfection on glass coverslips. At 48 h after transfection, cells were rinsed with PBS, and fixed with paraformadehyde 4% for 15 min. After PSB wash, cells were permeabilized and nonspecific protein binding blocked in 10% SVF and 0.1% Triton in PBS for 1 h at room temperature. Then, cells were incubated at room temperature for 1 h with primary antibodies diluted in blocking buffer (Table S4), rinsed with PBS, incubated 1 h at room temperature in dark in secondary antibody at a 1:1,000 dilution in blocking buffer (Alexa Fluor 647/555/488-conjugated goat anti-rabbit/mouse/rat). The actin cytoskeleton was stained with Alexa Fluor 647 or 488 Phalloidin (Invitrogen). Cell nuclei were stained by DAPI at 1 µg/ml in PBS for 10 min at room temperature. Coverslips were then mounted using ProLong Diamond (Molecular Probes), and imaged with 63× or 100× oil immersion objective of a wide-field microscope (DM RXA, Leica; camera CoolSNAP HQ, Photometrics), using Metamorph software.

## Data Availability

This study does not include data to be deposited in external repositories. The mass spectrometry data included in the source data associated to this article.

## Supplementary Information

## Acknowledgements

The authors are very grateful to Drs. C Pouponnot, M Alkobtawi, A Eychène, S Saule, and L Larue for insightful scientific discussions during this study and to Dr. V Petit for guidance in the culture of melanoma cell line. We deeply thank S Seal, C Pouponnot, and M Alkobtawi for their proofreading of the manuscript. We thank all the Monsoro-Burq team members for their constant support and Drs M Perron, P Gilardi, and C Pouponnot for acting as thesis committee advisors. Dr. A Eychène, Dr. S Druillenec, and Dr. L Larue kindly provided reagents and cell lines. We thank F Maczkowiak for mutagenesis of the *X. laevis pfkfb*4. We also acknowledge the valuable help from MN Soler and L Besse on Institut Curie Imaging platform facility PICT-IBiSA (Orsay), of C Lasgi for FACS analysis (Orsay), of L Barrio-Cano on Pasteur Institute Biomarkers platform facility (Paris), and members of the mass spectrometry platform at Jacques Monod Institute (Paris). We also acknowledge Dr. Staglar who provided the plasmid backbones and cell lines for the MaMTH experiment. This work was supported by grants to AH Monsoro-Burq from Université Paris Saclay, Centre National de la Recherche Scientifique (CNRS), Agence Nationale pour la Recherche (ANR-15-CE13-0012-01, ANR-21-CE13-0028-01), Fondation pour la Recherche Médicale (DEQ20150331733), and Institut Universitaire de France (IUF). M Sittewelle was supported by doctoral fellowships from Fondation Pour la Recherche Médicale (FRM ECO20160736105; FRM FDT201904007974). C Zhou was supported by IUF and ANR-21-CE13-0028-01 funding to AH Monsoro-Burq.

### Author Contributions

M Sittewelle: conceptualization, data curation, validation, investigation, visualization, methodology, and writing—original draft, review, and editing.
V Kappès: validation, investigation, visualization, methodology, and writing—review and editing.
C Zhou: validation, investigation, visualization, methodology, and writing—review and editing.
D Lècuyer: validation, investigation, visualization, and methodology.
AH Monsoro-Burq: conceptualization, supervision, funding acquisition, validation, investigation, visualization, project administration, and writing—original draft, review, and editing.

### Conflict of Interest Statement

The authors declare that they have no conflict of interest.

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
