## [Reviewer comments · Life Science Alliance]

Life Science Alliance

PFKFB4 interacts with ICMT and activates RAS/AKT signaling-dependent cell migration in melanoma.

Méghane Sittewelle, Vincent Kappès, Chenxi Zhou, Deborah Lécuyer, and Anne Monsoro-Burq

DOI: <https://doi.org/10.26508/lsa.202201377>

Corresponding author(s): Anne Monsoro-Burq, Institute Curie and Méghane Sittewelle, Institut Curie University Paris-Saclay

Review Timeline:

Submission Date:	2022-01-20
Editorial Decision:	2022-01-21
Revision Received:	2022-05-31
Editorial Decision:	2022-06-24
Revision Received:	2022-07-04
Accepted:	2022-07-06

Scientific Editor: Novella Guidi

Transaction Report:

Please note that the manuscript was reviewed at Review Commons and these reports were taken into account in the decision-making process at Life Science Alliance.

January 21, 2022

Re: Life Science Alliance manuscript #LSA-2022-01377

Prof. Anne-hélène Helene Monsoro-Burq
Institut Curie
Essonne
Centre Universitaire Batiment 110
orsay 91405
France

Dear Dr. Monsoro-Burq,

Thank you for submitting your manuscript entitled "The glycolysis regulator PFKFB4 interacts with ICMT and activates RAS/AKT signaling-dependent cell migration in melanoma." to Life Science Alliance. The manuscript was submitted and reviewed via Review Commons. The authors then chose to transfer their somewhat revised manuscript, along with the reviewers' comments and a proposed revised plan to Life Science Alliance (LSA). The reviewer comments and revision plan was assessed at LSA, and LSA editors deemed that the manuscript could be further considered at LSA provided the authors revise the manuscript, in accordance to what they have laid out in the pbp rebuttal / revision plan.

We, thus, encourage you to submit a revised manuscript to us that includes all the experiments you have laid out in their Revision plan. Given that new data will be added to the revised manuscript, the revision might have to be looked at by a set of referees, most likely the same ones as Review Commons.

Thank you for this interesting contribution to Life Science Alliance. We are looking forward to receiving your revised manuscript.

Sincerely,

B. MANUSCRIPT ORGANIZATION AND FORMATTING:

Full Revision

Manuscript number: RC-2021-00942

Corresponding author(s): Anne Hélène, Monsoro-Burq

[Please use this template only if the submitted manuscript should be considered by the affiliate journal as a full revision in response to the points raised by the reviewers.]

*If you wish to submit a preliminary revision with a revision plan, please use our "Revision Plan" template. **It is important to use the appropriate template to clearly inform the editors of your intentions.**]*

1. General Statements [optional]

This section is optional. Insert here any general statements you wish to make about the goal of the study or about the reviews.

Please find here our full revision for the above-mentioned study transferred to Life Science Alliance after reviewing by Review Commons. We have been in contact with Scientific Editor of Life Science Alliance, Dr. Novella Guidi, PhD, encouraging us to revise the manuscript, in accordance to what we had laid out in the rebuttal / revision plan.

In this study, we uncover a non-canonical function for the glycolysis regulator PFKFB4, affecting human metastatic melanoma cell migration. In this revised version, we further show that similar regulations affect non-cancerous melanocyte cells, and that this new function is not linked to significant alterations of glycolysis.

PFKFB1-4 enzymes are well known for their general role controlling glycolysis rate. PFKFB3 and PFKFB4 promote the Warburg effect in multiple cancer contexts. However, during development, we previously found that PFKFB4 displays non-conventional functions via AKT signaling, controlling early stem cells fate choices (Pegoraro et al., Nature Communications 2015) and neural crest cell migration (Figueiredo et al., Development 2017). Melanoma progression was recently shown to reactivate neural crest embryonic programs (Kaufman et al., Science 2016), and we found that both neural crest and melanoma cells express high levels of PFKFB4 (Plouhinec et al, PLOS Biology 2017).

Here, we find that PFKFB4 activates human melanoma cell migration, and that this function depends upon active AKT signaling. We further decipher the relationship between PFKFB4, AKT and cell migration. We first show that PFKFB4 activity differs from the effects of glycolysis and seems independent from PFKFB4 kinase/phosphatase functions. Instead, we identify ICMT as a direct partner of PFKFB4: ICMT is critical for RAS post-translational modification and localization at the plasma membrane, upstream of PI3K-AKT signaling. By a series of depletions and phenotype rescue experiments, we provide a new molecular mechanism affecting melanoma cell migration.

In revision, we have added the following experiments:

Full Revision

- New Fig 1C: western blot now includes the rescue experiment using non-mammalian (frog) PFKFB4: this validates the proper translation of this exogenous protein in human melanoma cells and the rescue as asked.
- Wound healing assay shows that PFKFB4 depletion affects melanoma cell healing properties (Fig. S2G);
- Seahorse experiments showing that cellular glycolysis parameters are not affected in melanoma cell lines after PFKFB4 depletion (Fig S3);
- Cell cycle and cell death analysis showing that are not affected in melanoma cell lines after PFKFB4 depletion (Fig. S4);
- PFKFB4 depletion in normal melanocytes, showing that they also respond to PFKFB4 by enhanced AKT signaling and migration (Fig S7).

This study thus shows that PFKFB4 displays a previously unsuspected function in melanoma biology affecting the control of cell migration via a novel PFKFB4-ICMT-RAS interaction. This new PFKFB4-based stimulation of melanoma cell migration is similar to neural crest migration mechanisms in embryos, and is also found in melanocytes. In melanoma, this mechanism occurs independently of the classically recorded melanoma driver mutations on RAS or downstream MAPK and PI3K-AKT signaling pathways. PFKFB4 could thus be a pivotal actor to target for therapeutic intervention in the future, especially in the triple-negative melanomas.

After the initial submission, all three reviewers reported that our findings were of broad interest for cancer biology and cell biology. In the attached point-to-point response to reviewers, we now have been able to address their comments in detail, including the new experiments mentioned above. We hope that in the present form, our study will be suitable for publication in Life Science Alliance.

Sincerely yours
Anne-Hélène Monsoro-Burq, Ph. D.

This section is mandatory. Please insert a point-by-point reply describing the revisions that were already carried out and included in the transferred manuscript.

1- Analysis of the cell's energy metabolism after depletion of PFKFB4.

Reviewer 1: Major comment 1:

The cells in the experiments are exposed to siRNA targeting PFKFB4 for 48 hrs with a demonstrated decrease in PFKFB4 protein. Their production of fructose-2, 6- BP and the effects of the treatment on glycolysis (by a more direct method than lactate production/growth in 2DG e.g. using tritiated glucose and measuring tritiated H₂O production) should be measured before concluding that glycolysis inhibition is not involved.

Reviewer 3: Major comment 3:

Full Revision

Cancer cells can display high metabolic flexibility and rewiring. What is the influence of siPFKFB4 on lactate and ATP production? Even slight decreases in energy could influence migration and not be further affected in low glucose conditions.

We thank reviewers 1 & 3 for this suggestion which prompted us to further explore the cells' metabolism after PFKFB4 depletion. In the initial version of the manuscript, as commonly done in the field, we have shown that blocking glycolysis directly using 2DG, an established glycolysis inhibitor, did not affect melanoma cells migration while depletion of PFKFB4 using siRNA targeting did. These distinct phenotypes suggested distinct underlying mechanisms. We now have directly assessed the effect of PFKFB4 depletion on the rate of glycolysis using lactate dosage as a glycolysis output parameter routinely monitored in melanoma-related and other cancer-related publications (e.g. <https://doi.org/10.1016/j.jid.2016.02.815>). In **novel Figure S3D**, we now show that lactate production is not significantly affected by PFKFB4 knockdown and modified the Result section lines 112-115 accordingly.

In addition, we are well aware that measuring lactate production is an indirect output of glycolysis efficiency (although used by most studies). Rather than using techniques involving radioactive compounds, to further validate the lack of correlation between glycolysis rate and cell migration phenotypes we performed a broad-spectrum glycolysis analysis on cells transfected with PFKFB4 siRNA using the Seahorse platform (Agilent) (see Material & methods section line 428-443). By measuring oxygen consumption rate (OCR) and extracellular acidification rate (ECAR) under various metabolism inhibitors (here rotenone/antimycin followed by 2-deoxyglucose), we deduced the global proton efflux rate (PER) and the PER associated only to the glycolysis (glycoPER) and were able to compare the PER, glycoPER and compensatory glycolysis in control and PFKFB4-depleted melanoma cells. None of these parameters showed a significant alteration after PFKFB4 knockdown (**novel Figure S3E-K**). We modified the Result section lines 115-124 accordingly.

These results are in agreement with the literature, where the relative dominant function of PFKFB4 (kinase or phosphatase) is debated and depends of the cellular context (Chesney et al 2014 DOI: 10.18632/oncotarget.2213; Sakata et al 1991 PMID: 1651918; Okar et al 2001 DOI: 10.1016/s0968-0004(00)01699-6, Ros et al 2012 DOI: 10.1158/2159-8290.CD-11-0234, Ros and Schulze 2013 DOI: 10.1186/2049-3002-1-8). Here, our results confirm that in our cell contexts, glycolysis rate and cell migration do not seem to be directly coupled.

2- Monitoring the effects of PFKFB4 depletion on cell death and cell cycle.

Reviewer 1: Major comment 2:

Effects on cell survival following siRNA treatment should be shown. Is the effect on migration in part due to cell death?

Reviewer 3: Major comment 1:

Full Revision

Changes in proliferation, cell cycle and viability can influence most of the parameters evaluated in the migration assay. How did knockdown of PFKFB4 influence proliferation? Presenting the timelapse videos and a more detailed assessment and discussion of this point would help.

Reviewer 4: Minor comment 7:

How was cell pausing defined? Were any live/dead staining or viability techniques used to ensure %pausing or decreased velocity was not due to cell death? Was this just determined by eye during monitoring? In general, was any viability difference observed after knockdown?

We thank the reviewers as these comments made us realize that a more detailed description of the cell migration tracking method was needed. We have now expanded the Material and Method section accordingly (lines 445-459): To assess the migration phenotype, we track single cells, plated at the same density in each condition. Randomly selected cells are first checked by eye to be alive over the entire duration of the movie. Moreover, we did not notice increased cell death in cells transfected by the siPFKFB4 compared to the control siRNA. The average speed is then calculated from the cell's trajectory (nucleus coordinates) over the duration of the movie. Pausing is the cumulative fraction of time of the total duration of the movie where the cells are not changing position. Cells pausing for mitosis were observed equally in all conditions. Thus, cell migration phenotypes seemed independent from cell growth and death processes. To support this description, we have now added an example of movie (**novel Supplementary material 1**) and clarified it in the Result section (lines 73-74).

In addition, to explicitly measure potential variations in cell cycle and in cell death, we have added now two series of new experiments: cells staining using 7-Aminoactinomycin D followed by fluorescence activated cell sorting (FACS) to measure cell viability and stoichiometric DNA staining also with 7-Aminoactinomycin D to measure the relative importance of each cell cycle phase (now explained in Materials and Methods lines 472 and following). No significant variation between the different conditions was observed as indicated in **novel Figure S4.**

3- Expression of xPFKFB4 in human melanoma cells and levels of expression in rescue experiment.

Reviewer 1: Major comment 5:

*In Figure 1C, the Western for PFKFB4 expression should include the effects of the *X. laevis* add back.*

Reviewer 4: Minor comment 1:

Figure 1C: The authors mention that their siRNA did not target the frog PFKFB4 and was specific to human. Did they try western blotting the frog xPFKFB4 with the human PFKFB4 antibody? Was it a similar enough sequence between the two genomes to give a signal to further validate rescue beyond the observed phenotypic velocity change?

Full Revision

This is an excellent point. We now have reproduced a full rescue experiment including the detection of xPFKFB4 by the antibody against the human protein to replace and improve the **Figure 1C**. This shows the proper translation of this exogenous protein in melanoma cells and validate the rescue as asked (lines 93-95).

4- Other comments from Reviewer 1

Reviewer 1 Major comment 3:

The concentration of siRNA used has not been provided in the methods.

We apologize for this omission. It is now added in the methods (line 368).

Reviewer 1 Major comment 4:

Lactate production should be normalized to cell number.

We now make it clear that the lactate readings were indeed normalized to the total protein content as directly proportional to the cell number. It is now clearly stated in the methods (line 425).

Reviewer 1: Major comment 6

Kinase and bisphosphatase mutants have been described- using point mutations designed for the liver isoform, PFKFB1 (based on the reference provided). Do these mutations inactivate kinase and phosphatase activity in PFKFB4? Activities should be measured to confirm.

We thank the reviewer for this comment that we would like to discuss in detail. The kinase and phosphatase domains of the four PFKFB1-4 isoenzymes is highly conserved at the amino-acid level (> 90%) as is the 3D protein structure (described in *Cancers* 2021, 13, 909. <https://doi.org/10.3390/cancers13040909>). Specifically, G48 and H258 are located in stretches of 100% conserved amino acids. Most cell biology studies, as ours here, thus rely on the deep conservation of the enzymatic domains either kinase or phosphatase for mutagenesis of the key residues. This is supported by the existing biochemistry analyses (e.g. quote from <https://doi.org/10.1016/j.jmb.2007.03.038>): “The observations made from this study are mostly in agreement with a number of previous functional studies that were performed on different PFKFB isoforms. This, together with the fact that constellations of the catalytic site residues in the active pocket are conserved among all PFKFB isoforms, suggests that the molecular catalytic mechanism derived from this study is applicable to all other PFKFB isoforms”).

Measuring the enzymatic activities of PFKFB proteins, in wild type and mutant conditions, would require protein production in large amounts, e.g. in bacterial cells, to perform kinase assays and phosphatase assays in vitro (e.g. <https://doi.org/10.1038/s41598-019-56708-0>). Moreover, the fructose 2,6 bisphosphate needed for the phosphatase assays is not commercially available. As our laboratory is specialized in cell and developmental biology, setting up such assays would be an entirely new project done through new collaborations, beyond the scope of a revision process.

As this point is not essential in the analysis of PFKFB4 function in cell migration and interaction with ICMT, we feel that addressing this comment by actual biochemical experiments

Full Revision

would drive us very far from the scope of this study for little benefit. We thus kindly request not to address this particular point.

Reviewer 1 Major comment 7:

The cells in Figure 3D and those in 3C appear to have been transfected similarly with activated HA-tagged Ras V12 and siPFKFB4. The RasV12 appears to rescue the migration phenotype in cells transfected with siPFKFB4 but does not similarly restore/increase cells with phenotype 1 in Figure 3D. The authors might discuss these findings.

We agree with the reviewer's observation. Our results suggest that in its constitutively active form, RasV12 may be efficiently rescuing the migratory phenotype even if it is found at the plasma membrane with low levels or "intermittent phenotype". We now mention this point in the Results and Discussion sections (lines 234-236, 288-291).

Reviewer 1 Major comment 8:

The effects of siPFKFB4 and siCMT on migration in Figure 3B are fairly different. The authors might discuss these findings.

We agree with the referee that the phenotypes observed on cell migration after depletion of PFKFB4 and ICMT do not reach exactly the same levels in all experiments, while they are *qualitatively* similar: phenotypes are quantitatively quite similar in the experiment shown Figure 3A (when compared to control, 0.598% average velocity for siPFKFB4 (40% of reduction) and 0.665% average velocity for siCMT (33% of reduction) and slightly different in Figure 3B (0.612% average velocity for siPFKFB4 (39% of reduction) and 0.779% average velocity reduction for siCMT (22% of reduction) compared to control).

However, this is due to the technical reproducibility between biological replicates. Moreover, it is always difficult to draw mechanistic conclusions from the quantitative comparison of two different siRNA. Significant variations can occur even when two siRNAs are designed against the same protein. Between two proteins, depletion efficiency can vary as do the endogenous protein quantity and its turnover. In addition, the minimum quantity of a protein needed in a cell to fully achieve its biological function is generally unknown. Last, ICMT and PFKFB4 are proteins with distinct additional targets, some of which potentially have effects on cell migration. We have assessed this point in the discussion (line 335): we state that PFKFB4 depletion affects the ICMT/RAS/AKT pathway which does not exclude that other PFKFB4 targets could participate in the migration phenotype. For instance, by mass spectrometry we also found PFKFB4 interaction with MMP14, a metalloprotease involved in cell migration and invasion: this could be studied in the future and explain some of the differences observed between PFKFB4 and ICMT.

4- Other comments from Reviewer 3

Reviewer 3: Major comment 2:

The differences in migration as indicated by quantification of random migration parameters, while statistically significant, seem modest even in representative experiments. Of course, even

Full Revision

a small difference could be biologically important. However, it makes interpretation of the effect of further perturbations even more difficult (for instance, figure 1G). I would suggest performing an additional migration assay for the main experiments.

We agree with the reviewer and we have been concerned by this point. We have already addressed it by repeating the experiment with two different siRNAs, more than 10 siRNA batches and a large number of biological replicates (22 biological replicates with MeWo and 5 with A375M). In fact, the average level of reduction of the migration efficiency is quite significant. In complete medium, the average reduction in velocity is of 33% for Mewo cells and of 42% for A375M cells compared to controls, over those 27 independent experiments. We have now added this information in the text (line 81) and in Figure 1 legend (line 724). In sum, this decrease is statistically significant, stable across experiments and cell lines, and of the same magnitude as those observed in other contexts (e.g. in prostate cancer cells doi: 10.18632/oncotarget.17821). Moreover, we have tracked cells by two independent means: either tracking all the cells in two to three fields randomly chosen (around 50 cells per condition) or tracking 10 cells randomly chosen in 10 different fields (a total of 100 cells per condition). Both strategies yielded the same results. We thus think that the PFKFB4 depletion phenotype would not be enhanced by additional biological replicates of the random single cell migration assays.

To further address the referee's concern, we have now also performed a wound healing scratch assay on MeWo cells transfected either with siCtrl or siPFKFB4 and monitored the gap closure over time (n=2 biological replicates – **new Figure S2-G,H**). This assay is commonly used to assess cells "migratory-like" phenotype, although we and other think that it rather assesses simultaneously cell proliferation and tissue healing. Nonetheless we also observed a significant decrease in wound closure efficiency after depletion of PFKFB4, confirming that PFKFB4 affects migration-related parameters as does the random single cell migration assay. Text, legends and Materials and Methods were altered accordingly (lines 83, 461, 799).

Reviewer 3: Minor comment 1:

Figure 1C should be mentioned in the text.

We apologize for this omission, it is now added to the text.

Reviewer 3: Minor comment 2:

Some of the axes in the figures use "um" instead of "µm" (Figure 1E, H, etc). Please check.

We thank the reviewer for this comment. It is now corrected in Figures and text.

Reviewer 3: Minor comment 3:

Is the FLAG band in Fig. 2B from a different blot?

The FLAG band in Figure 2B is from the same experiment, run on two PAGE gels migrated at the same time due to a lack of space on a single gel, transferred onto 2 different membranes simultaneously, and then processed together (incubated into the same antibody solution and revealed with the same ECL mix). Please see below the original membranes.

5- Other comments from Reviewer 4

Reviewer 4: Major comments:

The authors provide a convincing story outlining the basic characteristics of the PFKFB4-ICMT interaction and the role these two components likely play in melanoma cell migration. While there are some minor flaws within the paper and areas that could use additional support to make it even more impactful to the melanoma community, the authors do a good job in defining the scope of their work and the areas where it was lacking, which could provide future directions for this research.

We are very grateful to the reviewer for his time and for his positive comments about our work.

Reviewer 4: Minor text comments:

Consider revising minor grammatical errors and/or phrasing throughout as indicated by ():
 Abstract

- "In melanoma, oncogenic mutations frequently enhance RAS, PI3K/AKT or MAP kinase signaling, and trigger other cancer hallmarks(,) among which (includes) the activation of metabolism regulators."

Full Revision

Introduction

- *"Cell motility and invasiveness properties are reactivated during cancer progression(-) following the aberrant activation of multiple cellular programs, such as growth factor-independent signaling, metabolic and epigenetic reprogramming, which cooperate to sustain growth, proliferation, and survival properties in the primary tumor (3)."*
- *"Moreover, PFKFB4 is induced by hypoxia(- and) is required for survival and proliferation of normal thymocytes (21) (-) as well as of several cancer cell lines(.) such as lung, breast and colon adenocarcinomas, prostate and bladder cancer (22-25)."*
- *"For example, PFKFB4 regulates small (cell?) lung-cancer chemo-resistance by stimulating autophagy(-) via its interactions with Etk tyrosine kinase (31, 32)."*
- *"During development, PFKFB4 is essential for early embryonic induction(-) and neural crest cell(-) migration through the activation of AKT signaling (34, 35)."*

Materials and Methods

- *"PVDF" typographical error in western blotting section*

We thank the reviewer for these suggestions, it is now corrected.

Reviewer 4: Minor comment 2:

Figure 2D: Is there any way a cleaner blot could be produced for the RAS IP or is the background consistent among different runs?

Unfortunately, we could not get a stronger signal against immunoprecipitated endogenous RAS. Due to the low amount of RAS obtained after immunoprecipitation with ICMT (an integral RE membrane protein, which it is probably difficult maintain in a conformation still bound to RAS during immunoprecipitation), we are bound to use Femto ECL and long exposure which generates high background. However, we repeated the experiment successfully twice and reproduced the result.

Reviewer 4: Minor comment 3:

In the experiments that only demonstrate results in MeWo cells, were these also done in A375M cells? If not, what was the reasoning behind only doing one cell line after initially narrowing it down to two? If so, the authors may consider including these data in the supplemental section to further show it is similar (or dissimilar) across these two cell lines chosen, otherwise how is the reader to know this isn't specific to this one cell line?

Most of the experiments were done in at least two different cellular contexts. The A375M cell line is not as easy to manipulate as the MeWo cell line, notably for double transfections with siRNA and DNA. We initially have done all the experiments in the MeWo cell line and then

Full Revision

expanded the major assays/phenotypes with another cell line: 1) the migration of melanoma cells is diminished *in vitro* after PFKFB4 knockdown (MeWo and A375M) as is the migration of primary melanocytes (new Figure S7), 2) PFKFB4 and ICMT interact together (MeWo and HEK293), 3) Constitutive active RAS, a downstream known target of ICMT, can rescue the migration phenotype of depletion of PFKFB4 (MeWo and A375M), 4) The depletion of PFKFB4 alters the activating phosphorylation of AKT, a downstream effector of RAS (MeWo, MNT1, A375M and in normal melanocytes, but not in Lu1205), 5) constitutive active AKT can rescue the migration phenotype of depletion of PFKFB4 (MeWo and A375M).

In response to this referee, we now add the effects of PFKFB4 depletion on a non-cancer cell line, melanocytes (see minor comment 4 below, **new Figure S7**), strengthening the idea that this phenotype is broadly obtained in development (our previous analyses on neural crest), normal cell physiology (melanocytes) and melanoma.

Reviewer 4: Minor comment 4:

Discussion

"In sum, our study has demonstrated a novel, glycolysis-independent function of PFKFB4, promoting the interaction between ICMT and RAS, resulting in enhanced migration of melanoma cells (Figure 5)." The use of the term "enhanced" may not be the most appropriate here as low expression of PFKFB4 is not common in melanoma (as the authors have demonstrated earlier) and "enhanced" implies a comparison to another phenotype (which, in this case would not be one commonly seen in nature). It would be interesting to see how these experiments affect the phenotype of non-tumorigenic cells such as melanocytes as a kind of control.

We agree with the reviewer and have now changed for the more neutral term "active" migration (line 291). Moreover, we also have now analysed a melanocyte cell line, 12S2 mouse melanocytes, which also express PFKFB4 and migrate *in vitro*. We have validated siRNA for murine PFKFB4 and tested the phenotype on AKT phosphorylation and cell migration. Similarly to results for melanoma cells, we find that PFKFB4 impacts AKT signaling and cell migration efficiency (**new Figure S7**), new text, legend and materials and methods (lines 84, 257, 291, 359, 369, 861).

Reviewer 4: Minor comment 5:

The authors mention hypoxia a few times. Is it possible to include the comparison of hypoxic vs normoxic conditions across these experiments? If not, this may be included in the discussion as a further study (out of scope as mentioned with other experiments). Otherwise, it seems like an afterthought, only slightly related to the remainder of the study.

It seemed important to us to mention that PFKFB4 expression is known to be induced by hypoxia regarding the importance of this hallmark of cancer, notably concerning the Warburg effect, as many studies on PFKFB4 are based on one of these two aspect. The reviewer is right though, the comparison of hypoxic vs normoxic conditions is out of the scope of our study and

Full Revision

could be considered for a further study. We have now mentioned this aspect in discussion (lines 274).

Reviewer 4: Minor comment 6:

There were multiple antibodies listed within Table S4 which were not presented in any figure within the manuscript. Were these antibodies used and the data left out? Were these data significant? (PFKFB3, PTEN, PDK1, pPDK1, pGSK3b)

We apologize for this confusion and have removed these antibodies from the list. They were meant for another study and were not pursued here.

June 24, 2022

RE: Life Science Alliance Manuscript #LSA-2022-01377R

Prof. Anne Helene Monsoro-Burq
Institute Curie
Essonne
Centre Universitaire Batiment 110
Orsay 91405
France

Dear Dr. Monsoro-Burq,

Thank you for submitting your revised manuscript entitled "PFKFB4 interacts with ICMT and activates RAS/AKT signaling-dependent cell migration in melanoma.". We would be happy to publish your paper in Life Science Alliance pending final revisions necessary to meet our formatting guidelines.

- address Reviewer 2's remaining points
- please add an alternate abstract / summary blurb to our system
- please add the Twitter handle of your host institute/organization as well as your own or/and one of the authors in our system
- please consult our manuscript preparation guidelines <https://www.life-science-alliance.org/manuscript-prep> and make sure your manuscript sections are in the correct order and add a conflict of interest statement and the author contributions to the main manuscript text
- please use the [10 author names, et al.] format in your references (i.e. limit the author names to the first 10)
- we encourage you to introduce your panels in your figure legends in alphabetical order
- please double-check your callouts for Figure S5-you have callouts in the text that are not part of the figure or legend and seem to refer to Figure S6;
- please double-check your Figure S3 figure legend-you have callouts for figure S3K in the text, but this is not part of the legend
- please add a callout for Figure S6A-E and Figure S6H,I to your main manuscript text

Figure Check:

- please add sizes next to all blots
- please add scale bars to all microscopy images, and indicate size in figure legend

A. FINAL FILES:

-- Summary blurb (enter in submission system): A short text summarizing in a single sentence the study (max. 200 characters including spaces). This text is used in conjunction with the titles of papers, hence should be informative and complementary to the title. It should describe the context and significance of the findings for a general readership; it should be written in the

present tense and refer to the work in the third person. Author names should not be mentioned.

B. MANUSCRIPT ORGANIZATION AND FORMATTING:

Sincerely,

Reviewer #1 (Comments to the Authors (Required)):

1. This manuscript provides a more in-depth look at the role of PFKFB4 in melanoma cell migration, independent of its previously known function as a glycolysis regulator. The authors successfully demonstrate the independence of the stated functions of PFKFB4, its direct interaction with ICMT, and the effects of this interaction among RAS and PI3K/AKT signaling, leading to cell migration changes in vitro. Overall, the authors provide sound conclusions based on the data presented and make note of future directions and pitfalls to what has been proposed in this manuscript. These results surrounding the migration characteristics of the PFKFB4-ICMT interaction provide exciting insight into potential metastasis drivers.

2. The authors effectively introduced additional data in response to reviewer suggestions and successfully supported their experimental conclusions.

3. No additional concerns are of note.

Reviewer #2 (Comments to the Authors (Required)):

This revised manuscript from Sittewelle et al explores a novel function for PFKFB4 in melanoma cell migration. The authors present data finding that PFKFB4 interacts with ICMT and may promote the interaction of ICMT and RAS. They postulate that this interaction may control RAS localization to the plasma membrane to activate AKT signaling and enhance cell migration and that these data provide evidence of a novel glycolysis-independent function for PFKFB4 in melanoma.

The data are certainly interesting and significant in that they postulate a previously undescribed role for PFKFB4 in cell migration. The authors have been responsive to reviewer comments and the revisions made have significantly improved the manuscript. There however remain a couple of unresolved questions that need to be addressed.

The authors have provided data indicating that in melanoma cells, glucose deprivation does not appear to affect the process of

migration and that glycolysis appears to be similar in control and PFKFB4 knockdown cells. They have not shown results of ATP determination in the cells and whether PFKFB4 silencing affects cellular ATP concentration. If not through glycolysis, what process are the cells deriving ATP from to fuel migration and growth? Did the Seahorse data find that the cells have increased respiration? It would be pertinent to show/discuss these results.

The authors have now included data indicating that primary immortal melanocytes also rely on PFKFB4 for migration. It is unclear what kind of mice 12S2 cells are derived from. Do these cells express RAS? If not, what mechanism are they using? Were there differences in migration between the primary and transformed cells or do the authors postulate that this is a mechanism common to all melanocytes and not specific to cancer cells? These issues should be addressed.

In the data shown in Figure 4D, pAKT appears to be higher in the 2 and 5mM 2DG treated cells relative to control. The authors should address this finding.

July 6, 2022

RE: Life Science Alliance Manuscript #LSA-2022-01377RR

Prof. Anne Helene Monsoro-Burq
Institute Curie
Essonne
Centre Universitaire Batiment 110
Orsay 91405
France

Dear Dr. Monsoro-Burq,

Thank you for submitting your Research Article entitled "PFKFB4 interacts with ICMT and activates RAS/AKT signaling-dependent cell migration in melanoma.". It is a pleasure to let you know that your manuscript is now accepted for publication in Life Science Alliance. Congratulations on this interesting work.

DISTRIBUTION OF MATERIALS:

Again, congratulations on a very nice paper. I hope you found the review process to be constructive and are pleased with how the manuscript was handled editorially. We look forward to future exciting submissions from your lab.

Sincerely,
